# Analytical Solution for Electromagnetic Performance Analysis of Permanent Magnet Synchronous Motor with a Parallel Magnetized Cylindrical Permanent Magnet

**Hao Lin** [1,*], **Haipeng Geng** [2], **Ling Li** [1], **Leiming Song** [1] **and Xiaojun Hu** [1]

1   School of Mechanical, Electronic and Control Engineering, Beijing Jiaotong University, Beijing 100044, China; 23111369@bjtu.edu.cn (L.L.); lmsong@bjtu.edu.cn (L.S.); xjhu@bjtu.edu.cn (X.H.)
2   School of Mechanical Engineering, Xi'an Jiaotong University, Xi'an 710049, China; genghaipeng@xjtu.edu.cn
*   Correspondence: haol@bjtu.edu.cn

**Abstract:** High-speed direct-drive permanent magnet synchronous motors (PMSMs), supported by elastic foil gas bearings, have broad applications, such as in microcompressors. However, some problems remain to be solved for the electrical performance analysis of PMSMs. For example, there is presently no related analytical model that can be used in rotor dynamics expression for this type of PMSM. This study aimed to establish theoretical models for electromagnetic force density and torque. The process involved both theoretical and experimental research. The analytic models of air gap magnetic density, electromagnetic force density, and electromagnetic performance were established for a PMSM with a parallel magnetized cylindrical permanent magnet. The analytic calculation was conducted, and the results of the analytic model were obtained. The analytical model of the electromagnetic torque and force can be applied in theoretical research on rotor dynamics. The model provides a theoretical basis and method for studying the influence of the electromagnetic load on rotor dynamics. A finite element simulation analysis of the electrical performance of the PMSM was carried out. An electrical performance experiment was conducted. The deviation between the experimental result and the theoretical value was less than 4%. This result indicated that the analytic models could be used in a dynamics analysis of compressors that are directly driven by a PMSM for application in engineering and industrial contexts.

**Keywords:** analytical solution; electromagnetic force density; electromagnetic performance; permanent magnet synchronous motor; parallel magnetized cylindrical permanent magnet

## 1. Introduction

High-speed direct-drive permanent magnet synchronous motors (PMSMs), supported by elastic foil gas bearings, have many applications. Their power range is from 10 to 200 kW, and their speed range is from 30 to 100 kr/min (the unit kr/min refers to 1000 revolutions per minute). Compressors that are directly driven by high-speed PMSMs have a broad range of applications [1].

High-speed PMSM-driven technologies offer advantages: compact structure, light weight, and high efficiency. High-speed direct-drive mechanisms are key technologies in this field [2,3]. High-speed direct-drive PMSMs can be widely used because they have the following attributes: (1) the coaxial-drive mode makes the system structure more compact, reducing mechanical loss by about 15% through the omission of the gearbox [4]; (2) it can obtain high power density at high speeds, significantly reducing the volume and weight [5]; (3) the efficiency of PMSMs is more than 10% higher than that of traditional induction motors [6]. Therefore, the research and development of high-speed direct-drive PMSMs has become essential in the development of high-end energy power equipment.

Electromagnetic analyses of high-speed PMSMs obey basic electromagnetic principles. However, the impact of the electromagnetic field on the permanent magnet rotor needs



to be analyzed due to the following characteristics: high speed, parallel magnetized cylindrical permanent magnet, and elastic foil bearing support. Analyses of electromagnetic fields generally adopt two methods: analytical analyses and simulation analyses. Many researchers have provided simulation results from electromagnetic fields through finite element analysis [7,8]. Finite element simulations are necessary for guaranteeing PMSM performance in various areas, such as the occurrence of stator saturation; meanwhile, analytical models can be used to determine the influence of the electromagnetic field on rotor vibration. Because electromagnetic loads affect the vibration of the rotor through the air gap magnetic field, converting the analytic expression of the electromagnetic field into that of the electromagnetic load is crucial, as it can be included in the dynamic equation of the rotor. The intention of this study is to find an analytic solution that can be beneficial for further mechanical modeling and analysis of PMSM rotor dynamics.

The diversity in the shapes and assembly methods of permanent magnets means that analytical models of their magnetic fields are not universally acceptable. Various studies have provided methodologies for analytically determining these magnetic fields. For multipole surface-mounted permanent magnets, the magnetic circuit analysis method has been obtained based on a 2D magnetic field [9]. The magnetic field control equation in a multipole slot-less annular air gap region was solved through the polar coordinate system through a 2D analytical method for magnetic field distribution. Assuming that a permanent magnet has a uniform radial magnetization and a constant relative permeability, this solution provides good versatility [10]. The analytical method of ring harmonics was used to calculate the 3D magnetic field of parallel magnetized cylindrical permanent magnets [11]. The authors of both [12,13] provided analytical expressions for the radial and tangential magnetic densities for permanent magnets.

In PMSMs with a parallel magnetized cylindrical permanent magnet, the air gap magnetic field results from the superimposition of the magnetic field of the permanent magnet and the armature under the load state. The analytical model for the armature magnetic field of traditional induction motors was established based on a 1D model in the Cartesian coordinate system [14]. It had an excessive equivalent air gap for PMSMs, with significant deviation in the results. The analytical solution for the three-phase armature magnetic field of the brushless DC motor was obtained using the equivalent current sheet. The analytical model considered the large effective air gap and the influence of winding the current harmonics on the air gap magnetic field [15]. The authors of [16] provided a harmonic analytical solution for the armature magnetic field in a polar coordinate system. Previous studies [17,18] have established an analytical model for the armature magnetic field of PMSMs.

The electromagnetic load includes electromagnetic force and electromagnetic torque. Analytical expressions have been based on the densities of the radial and tangential electromagnetic forces. The analytical expression of the radial electromagnetic force density could be given based on the air gap magnetic density. Kim [19] examined the calculation of radial electromagnetic force density for interior permanent magnet (IPM) and surface-mounted permanent magnet (SPM) motors. Dorrell [20] explored the radial electromagnetic force density of brushless motors. Previous studies [21,22] have shown that the analytical model of the radial electromagnetic force density was obtained using the stress tensor method based on the magnetic density. In addition, the radial and tangential components of air gap magnetic density were used to derive the radial electromagnetic force density of a two-pole parallel magnetized cylindrical permanent magnet. Therefore, this study established the analytical expression of the air gap magnetic density and electromagnetic force density for PMSMs with a parallel magnetized cylindrical permanent magnet.

The solution methods for electromagnetic torque are not similar for different types of motors. Some studies provided analytical models for electromagnetic torque. Bermúdez [23] obtained the analytical model of electromagnetic torque by the Maxwell stress tensor method. Lundström [24] calculated the electromagnetic torque of the hydroelectric generator. Dorrell [25] proposed the analytical method for the electromagnetic torque of

squirrel-cage asynchronous motors. Guo [26] theoretically derived the electromagnetic torque model of a three-phase generator in the no-load condition using harmonic analysis methods. In summary, the aforementioned research methods were used for different types of motors, but no analytical expression was obtained for the electromagnetic torque of two-pole parallel magnetized PMSMs.

This study aimed to establish theoretical models of electromagnetic force density and torque for PMSMs with a parallel magnetized cylindrical permanent magnet in Section 2. There was presently no related analytical expression that can be applied in the rotor dynamics expression for this type of PMSM. This was different in the proposed model than in existing models. Then, analytical calculations were conducted for air gap magnetic density, electromagnetic force density and electromagnetic torque of PMSMs in Section 3. Furthermore, in Section 4, operational experiments were carried out using power output data to verify the analytical results. This study has the following contributions. First, this analytical model of electromagnetic torque and force, which is the load in the dynamic equation, can be applied in the theoretical research on rotor dynamics. It provides the theoretical basis and method for studying the influence of electromagnetic load on rotor dynamics. Second, analytic results can be used as references for torque and power in analyzing the electromagnetic performance of compressors that are directly driven by PMSMs.

## 2. Theoretical Model

### 2.1. Air Gap Magnetic Field

In this study, some assumptions were made to simplify calculation: (1) the end effect was ignored, i.e., the axial length of the motor was regarded as infinite using the 2D model; (2) the magnetic permeability of the stator core was extremely large compared with the air, so it could be regarded as infinite in the model; (3) the permanent magnet had a linear demagnetization curve; (4) the material properties were linear and isotropic.

In the absence of current in the stator winding, the permanent magnet generated an air gap magnetic field, as shown in Figure 1. Due to the high speed of PMSMs, it was necessary to minimize the number of permanent magnet poles as much as possible. Therefore, the permanent magnet had one pair of poles using parallel magnetization.

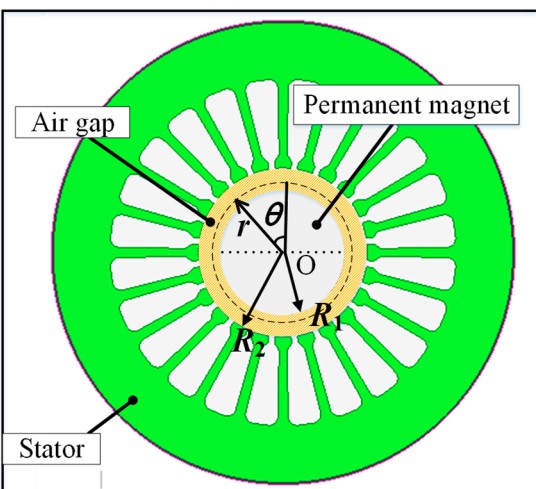

**Figure 1.** Schematic diagram of air gap magnetic field [13].

$R_1$ and $R_2$ represented the radius of the permanent magnet and the inner radius of the stator, respectively. The air gap area was defined as a nonmagnetic material. In Figure 1, $r$ represented the radius of the air gap magnetic field, and $\theta$ represented the angle of the air gap magnetic field. The range of $r$ was from $R_1$ to $R_2$.

All symbols in Equations (1)–(6) were explained in Table 1.

**Table 1.** Nomenclature of symbols in Equations (1)–(6).

| Symbol | Description | Symbol | Description |
|---|---|---|---|
| $B_r^P$ | radial component of the magnetic flux density generated by the permanent magnet | $B_\theta^P$ | tangential component of the magnetic flux density generated by the permanent magnet |
| $B_{r1}^P$ | amplitude of the radial magnetic flux density generated by the permanent magnet | $B_{\theta 1}^P$ | amplitude of the tangential magnetic flux density generated by the permanent magnet |
| $r$ | radius of the air gap magnetic field | $\theta$ | angle of the air gap magnetic field |
| $R_1$ | radius of the permanent magnet | $R_2$ | inner radius of the stator |
| $\mu_0$ | vacuum permeability | $\mu_r$ | relative permeability of the permanent magnet |
| $B_r^W$ | radial component of the magnetic flux density generated by the armature | $B_\theta^W$ | tangential component of the magnetic flux density generated by the armature |
| $B_r^{phA/B/C}$ | radial component of the magnetic flux density generated by the A, B and C phase winding | $B_\theta^{phA/B/C}$ | tangential component of the magnetic flux density generated by the A, B and C phase winding |
| $q$ | number of slots per pole and per phase | $N_s$ | number of conductors per slot |
| $K_{dp\nu}$ | winding coefficient of the $\nu$-order harmonic | $K_{so\nu}$ | slot opening coefficient of the $\nu$-order harmonic |
| $i_a, i_b, i_c$ | current of A-phase, B-phase and C-phase | $M_R$ | magnetization intensity |
| $B_r^L$ | radial component of the load magnetic flux density | $B_\theta^L$ | tangential component of the load magnetic flux density |
| $I_m$ | amplitude of the current | $\omega_0$ | angular velocity |
| $\beta$ | phase angle of the current | $\gamma$ | angle between A-phase winding and $d$-axis |

The analytical formula for the magnetic flux density of the permanent magnet with parallel magnetizing was as follows:

$$\begin{cases} B_r^P(r,\theta) = B_{r1}^P \cos\theta \\ B_\theta^P(r,\theta) = B_{\theta 1}^P \sin\theta \end{cases}, \tag{1}$$

where amplitudes of the radial and tangential magnetic fields were [13]:

$$\begin{cases} B_{r1}^P(r) = \dfrac{-\mu_0 M_R R_1^2}{(\mu_r-1)R_1^2-(\mu_r+1)R_2^2}(1+R_2^2 r^{-2}) \\ B_{\theta 1}^P(r) = \dfrac{-\mu_0 M_R R_1^2}{(\mu_r-1)R_1^2-(\mu_r+1)R_2^2}(-1+R_2^2 r^{-2}) \end{cases}, \tag{2}$$

where $\mu_0$ is the vacuum permeability; $\mu_r$ is the relative permeability of the permanent magnet; $M_R$ is the magnetization intensity.

In the three-phase winding coordinate system, the armature magnetic field of the three-phase winding was [17]:

$$\begin{cases} B_r^W & = B_r^{phA} + B_r^{phB} + B_r^{phC} \\ & = \dfrac{-2qN_s\mu_0}{\pi}\sum\limits_{v=1}^{\infty}\left\{\left[K_{dpv}(2v-1)K_{sov}(2v-1)\cdot F_0(2v-1,r)(-1)^v\right]\right. \\ & \left. \times[i_a\cos(2v-1)\theta + i_b\cos(2v-1)(\theta-\tfrac{2\pi}{3}) + i_c\cos(2v-1)(\theta-\tfrac{4\pi}{3})]\right\} \\ B_\theta^W & = B_\theta^{phA} + B_\theta^{phB} + B_\theta^{phC} \\ & = \dfrac{-2qN_s\mu_0}{\pi}\sum\limits_{v=1}^{\infty}\left\{\left[K_{dpv}(2v-1)K_{sov}(2v-1)\cdot F_0(2v-1,r)(-1)^{v-1}\right]\right. \\ & \left. \times[i_a\sin(2v-1)\theta + i_b\sin(2v-1)(\theta-\tfrac{2\pi}{3}) + i_c\sin[(2v-1)(\theta-\tfrac{4\pi}{3})]\right\} \end{cases} \tag{3}$$

where $B_r^{phA}$, $B_r^{phB}$ and $B_r^{phC}$ are the radial components of the magnetic flux density generated by the A-phase, B-phase and C-phase windings, respectively; $B_\theta^{phA}$, $B_\theta^{phB}$ and $B_\theta^{phC}$ are

tangential components of the magnetic flux density generated by the A-phase, B-phase and C-phase windings, respectively; $q$ is the number of slots per pole and per phase; $N_s$ is the number of conductors per slot; $K_{dp\nu}$ is the winding coefficient of the $\nu$-order harmonic; $K_{so\nu}$ is the slot opening coefficient of the $\nu$-order harmonic; $i_a$, $i_b$ and $i_c$ are the current of the A-phase, B-phase and C-phase windings, respectively.

The coordinate system of the permanent magnet magnetic field was along the magnetizing direction (*d*-axis), and that of the armature magnetic field was along the axis direction of the A-phase winding. When calculating the magnetic density of the air gap, combining two coordinate systems was necessary, as shown in Figure 2. The angle between the axis of the A-phase winding and the *d*-axis of the permanent magnet was $\gamma$.

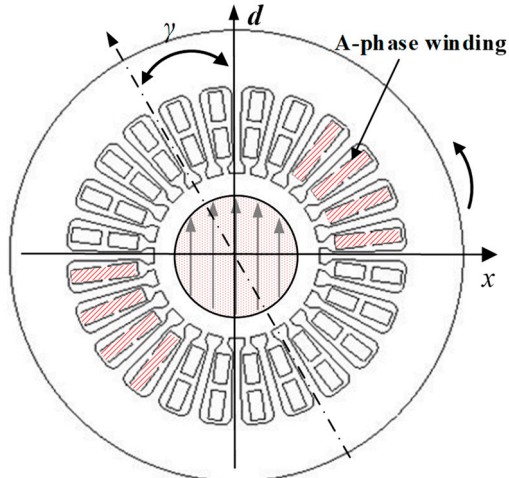

**Figure 2.** Schematic diagram of angle between A-phase winding and *d*-axis (two-pole motor) [13].

Based on the magnetic field analytical expressions of the permanent magnet and armature, air gap magnetic density was superposed according to the following formulas [27]:

$$\begin{cases} B_{rx}^{L}(t,r,\varphi) = B_{rx}^{W} + B_{rx}^{P} \\ B_{\theta x}^{L}(t,r,\varphi) = B_{\theta x}^{W} + B_{\theta x}^{P} \end{cases}, \tag{4}$$

where the superscripts L, W, and P represent the magnetic field of load, armature, and permanent magnet, respectively; the subscript x represents the variable in the coordinate system of the *x*-axis; *t* represents the time, and $\varphi = \theta - \pi/2$.

The analytical expressions of the air gap magnetic density could be obtained as follows:

$$B_{rx}^{L}(t,r,\varphi) = \frac{-2qN_sI_m\mu_0}{\pi} \sum_{v=1}^{\infty} \left\{ \left[ K_{dpv}(2v-1)K_{sov}(2v-1)F_0(2v-1,r)(-1)^v \right] \right.$$
$$\times \left[ \cos(\omega_0 t + \beta)\cos(2v-1)(\varphi-\gamma) + \cos(\omega_0 t + \beta - \tfrac{2\pi}{3})\cos(2v-1)(\varphi-\gamma-\tfrac{2\pi}{3}) \right.$$
$$\left. + \cos(\omega_0 t + \beta - \tfrac{4\pi}{3})\cos(2v-1)(\varphi-\gamma-\tfrac{4\pi}{3})] \right\}$$
$$+ \frac{-\mu_0 M_R R_1^2(1+R_2^2 r^{-2})}{(\mu_r-1)R_1^2-(\mu_r+1)R_2^2} \cos(\varphi-\omega_0 t) \tag{5}$$

$$B_{\theta x}^{L}(t,r,\varphi) = \frac{-2qN_sI_m\mu_0}{\pi} \sum_{v=1}^{\infty} \left\{ \left[ K_{dpv}(2v-1)K_{sov}(2v-1)F_0(2v-1,r)(-1)^{v-1} \right] \right.$$
$$\times \left[ \cos(\omega_0 t + \beta)\sin(2v-1)(\varphi-\gamma) + \cos(\omega_0 t + \beta - \tfrac{2\pi}{3})\sin(2v-1)(\varphi-\gamma-\tfrac{2\pi}{3}) \right.$$
$$\left. + \cos(\omega_0 t + \beta - \tfrac{4\pi}{3})\sin[(2v-1)(\varphi-\gamma-\tfrac{4\pi}{3})]] \right\}$$
$$+ \frac{-\mu_0 M_R R_1^2(-1+R_2^2 r^{-2})}{(\mu_r-1)R_1^2-(\mu_r+1)R_2^2} \sin(\varphi-\omega_0 t) \tag{6}$$

where $I_m$ is the amplitude of the current; $\beta$ is the phase angle of the current; $\omega_0$ is the angular velocity.

### 2.2. Electromagnetic Force Density

The radial and tangential components of the air gap magnetic density were used to derive the radial and tangential electromagnetic force density. Therefore, the analytical expressions of the electromagnetic force density for PMSMs with a parallel magnetized cylindrical permanent magnet were established as mentioned below.

Both the electromagnetic force and torque of PMSMs could be regarded as the result of the interaction between the magnetic field of the permanent magnet and the stator. Electromagnetic torque could be expressed as the integral of the tangential electromagnetic force density on the surface of the permanent magnet.

All symbols in Equations (7)–(14) were explained in Table 2.

**Table 2.** Nomenclature of symbols in Equations (7)–(14).

| Symbol | Description | Symbol | Description |
|---|---|---|---|
| $f_r$ | radial electromagnetic force density | $f_\theta$ | tangential electromagnetic force density |
| $\bar{f}_{rx}^L$ | stationary component of the radial electromagnetic force density | $\widetilde{f}_{rx}^L$ | harmonic component of the radial electromagnetic force density |
| $\bar{f}_{\theta x}^L$ | stationary component of the tangential electromagnetic force density | $\widetilde{f}_{\theta x}^L$ | harmonic component of the tangential electromagnetic force density |

Using the Maxwell stress tensor method, the radial and tangential electromagnetic force density acting on the permanent magnet surface were

$$\begin{cases} f_{rx}^L(t,r,\varphi) = \frac{(B_{rx}^L(t,r,\varphi))^2 - (B_{\theta x}^L(t,r,\varphi))^2}{2\mu_0} \\ f_{\theta x}^L(t,r,\varphi) = \frac{B_{rx}^L(t,r,\varphi)B_{\theta x}^L(t,r,\varphi)}{\mu_0} \end{cases}. \tag{7}$$

According to Equations (5) and (6), when $\nu = 1$ and $r = R_1$, the radial electromagnetic force density on the surface of the permanent magnet was as follows:

$$\begin{aligned} f_{rx}^L(t,\varphi) = \quad & \mu_0 C_{rx}^1 D_{x1}^{PM} \cos\delta + \mu_0 D_{x1}^{PM} D_{x2}^{PM} R_1^{-2} + \frac{1}{2}\mu_0 (C_{rx}^1)^2 \cos 2(\omega_0 t - \varphi + \delta) \\ & + \mu_0 C_{rx}^1 D_{x2}^{PM} R_1^{-2} \cos(2(\omega_0 t - \varphi) + \delta) \\ & + \frac{1}{2}\mu_0 [(D_{x1}^{PM})^2 + (D_{x2}^{PM} R_1^{-2})^2] \cos 2(\varphi - \omega_0 t) \end{aligned} \tag{8}$$

where $\delta = \beta + \gamma$. $\delta$ represents the included angle between the axis of the armature magnetic field and the axis of the permanent magnet magnetic field.

The constant terms were as follows:

$$\begin{cases} C_{rx}^1 = \frac{3q N_s K_{dp\nu}(1) K_{so\nu}(1) I_m}{\pi R_2} \\ D_{x1}^{PM} = \frac{M_R R_1^2}{R_2^2(\mu_r+1) - R_1^2(\mu_r-1)} \\ D_{x2}^{PM} = \frac{M_R R_1^2 R_2^2}{R_2^2(\mu_r+1) - R_1^2(\mu_r-1)} \end{cases} \tag{9}$$

In Equation (8), the radial electromagnetic force density could be divided into two parts. One part was as follows:

$$\bar{f}_{rx}^L = \mu_0 C_{rx}^1 D_{x1}^{PM} \cos\delta + \mu_0 D_{x1}^{PM} D_{x2}^{PM} R_1^{-2} \tag{10}$$

This part was a stationary component. The first item represented the interaction of the magnetic field of the armature and the permanent magnet. The second item represented the magnetic field of the permanent magnet.

The second-order harmonic component was as follows:

$$
\begin{aligned}
\widetilde{f}_{\mathrm{rx}}^{\mathrm{L}}(t,\varphi) =\ & \tfrac{1}{2}\mu_0(C_{\mathrm{rx}}^1)^2\cos 2(\omega_0 t - \varphi + \delta) + \mu_0 C_{\mathrm{rx}}^1 D_{\mathrm{x2}}^{\mathrm{PM}} R_1^{-2}\cos(2(\omega_0 t - \varphi) + \delta) \\
& + \tfrac{1}{2}\mu_0[(D_{\mathrm{x1}}^{\mathrm{PM}})^2 + (D_{\mathrm{x2}}^{\mathrm{PM}} R_1^{-2})^2]\cos 2(\varphi - \omega_0 t)
\end{aligned}
\tag{11}
$$

The first item reflected the armature magnetic field. The third item reflected the permanent magnet magnetic field. The second item reflected the coupling effect of the armature and the permanent magnet.

Similarly, when $\nu = 1$, according to Equation (7), the tangential electromagnetic force density at the outer surface of the permanent magnet was obtained as:

$$
\begin{aligned}
f_{\theta\mathrm{x}}^{\mathrm{L}}(t,\varphi) =\ & \mu_0 C_{\mathrm{rx}}^1 D_{\mathrm{x1}}^{\mathrm{PM}}\sin\delta \\
& + \tfrac{1}{2}\mu_0[(C_{\mathrm{rx}}^1)^2\sin 2\delta + 2C_{\mathrm{rx}}^1 D_{\mathrm{x2}}^{\mathrm{PM}} R_2^{-2}\sin\delta]\cos 2(\omega_0 t - \varphi) \\
& + \tfrac{1}{2}\mu_0[(C_{\mathrm{rx}}^1)^2\cos 2\delta + 2C_{\mathrm{rx}}^1 D_{\mathrm{x2}}^{\mathrm{PM}} R_2^{-2}\cos\delta \\
& - ((D_{\mathrm{x1}}^{\mathrm{PM}})^2 - (D_{\mathrm{x2}}^{\mathrm{PM}} R_2^{-2})^2)]\sin 2(\omega_0 t - \varphi)
\end{aligned}
\tag{12}
$$

The tangential electromagnetic force density acting on the outer surface of the permanent magnet also contained two parts.

The stationary component determined the electromagnetic torque to maintain the constant rotor angular velocity $\omega_0$, and was as follows:

$$
\overline{f}_{\theta\mathrm{x}}^{\mathrm{L}} = \mu_0 C_{\mathrm{rx}}^1 D_{\mathrm{x1}}^{\mathrm{PM}}\sin\delta.
\tag{13}
$$

The expression of the second harmonic component reflected the interaction between the armature and the permanent magnet, including the second sine wave and the second cosine wave. The second-order harmonic component was as follows:

$$
\begin{aligned}
\widetilde{f}_{\theta\mathrm{x}}^{\mathrm{L}}(t,\varphi) =\ & \tfrac{1}{2}\mu_0[(C_{\mathrm{rx}}^1)^2\sin 2\delta + 2C_{\mathrm{rx}}^1 D_{\mathrm{x2}}^{\mathrm{PM}} R_2^{-2}\sin\delta]\cos 2(\omega_0 t - \varphi) \\
& + \tfrac{1}{2}\mu_0[(C_{\mathrm{rx}}^1)^2\cos 2\delta + 2C_{\mathrm{rx}}^1 D_{\mathrm{x2}}^{\mathrm{PM}} R_2^{-2}\cos\delta \\
& - ((D_{\mathrm{x1}}^{\mathrm{PM}})^2 - (D_{\mathrm{x2}}^{\mathrm{PM}} R_2^{-2})^2)]\sin 2(\omega_0 t - \varphi)
\end{aligned}
\tag{14}
$$

### 2.3. Electromagnetic Load

In the analysis process of high-speed PMSMs, considering the influence of compressor power load on the PMSM dynamic electromagnetic performance is crucial, mainly including the electromagnetic torque.

The electromagnetic torque of PMSMs was transmitted through the air gap magnetic field between the stator and the rotor. The electromagnetic torque acting on the rotor could be regarded as the interaction between the permanent magnet and the armature. Based on Equation (13), electromagnetic torque was obtained by integrating the stationary component as follows:

$$
T_{\mathrm{em}} = \int_{\theta_1}^{\theta_2}\overline{f}_{\theta\mathrm{x}}^{\mathrm{L}}(R_1,\theta)R_1^2 L_m d\theta,
\tag{15}
$$

where $L_{\mathrm{m}}$ is the axial length of the rotor permanent magnet; $\theta_1$ and $\theta_2$ were the starting position angle and the ending position angle, respectively, and $\theta_1 = 0$ and $\theta_2 = 2\pi$.

Substituting Equation (13) into Equation (15), the following expression was obtained:

$$
T_{em}(\delta) = 2\pi\mu_0 R_1^2 L_{\mathrm{m}} C_{\mathrm{rx}}^1 D_{\mathrm{x1}}^{\mathrm{PM}}\sin\delta,
\tag{16}
$$

The expression of electromagnetic power was as follows:

$$
P_{em}(\delta) = T_{em}\omega_0 = 2\pi\mu_0 R_1^2 L_{\mathrm{m}}\omega_0 C_{\mathrm{rx}}^1 D_{\mathrm{x1}}^{\mathrm{PM}}\sin\delta,
\tag{17}
$$

The relationship between the electromagnetic power of the PMSM and the load power of the compressor satisfied the following expression:

$$P_0 = k_e \eta_d \eta_e P_{em}, \tag{18}$$

where $P_{em}$ is the electromagnetic power of the PMSM; $P_0$ is the load power of the compressor; $\eta_d$ is the transmission efficiency; and $\eta_e$ is the compression efficiency reflecting the friction loss and compressed gas loss. $k_e$ is the margin between the electromagnetic power and the theoretical load power of the compressor.

Under the action of electromagnetic torque and load torque, the rotor torsional vibration equation was as follows:

$$J_z \frac{d\omega_m}{dt} + B\omega_m = T_{em} + T_1, \tag{19}$$

where $J_z$ is the rotational inertia of the rotor around the Z axis; $\omega_m$ is the mechanical rotation angular speed of the rotor; $B\omega_m$ is the friction damping term; $T_1$ is load torque; and $T_{em}$ is the electromagnetic torque applied to the rotor.

## 3. Analytical Calculation

The structural parameters, which were required to set up the analytic model of PMSMs, are given in Table 3.

**Table 3.** Structural parameters of PMSMs.

| Parameter | Value | Parameter | Value |
|---|---|---|---|
| Permanent magnet length $L_m$ (mm) | 32 | Outer radius of permanent magnet $R_1$ (mm) | 13.75 |
| Inner radius of stator $R_2$ (mm) | 19 | Number of slots $N_s$ | 24 |
| Slot width $b_0$ (mm) | 1.5 | Electrical angle of slot pitch $\alpha_t$ | $\pi/12$ |

By analyzing the PMSM in Table 3, the air gap magnetic density along the inner surface of the stator and the outer surface of the permanent magnet under the load condition was obtained using Equations (5) and (6), as shown in Figures 3 and 4, respectively. The time coordinate axis represented the variation in magnetic density with time (three electric cycles). The angular coordinate axis represented the variation in magnetic density with the air gap angle (one mechanical cycle).

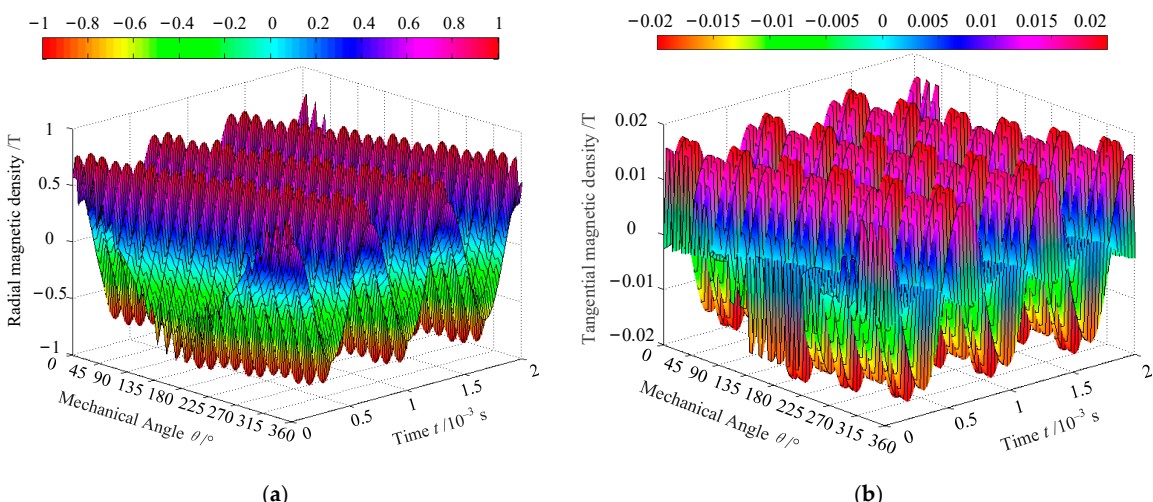

(**a**)    (**b**)

**Figure 3.** Distribution of magnetic density along the inner surface of the stator: (**a**) Radial magnetic density; (**b**) Tangential magnetic density.

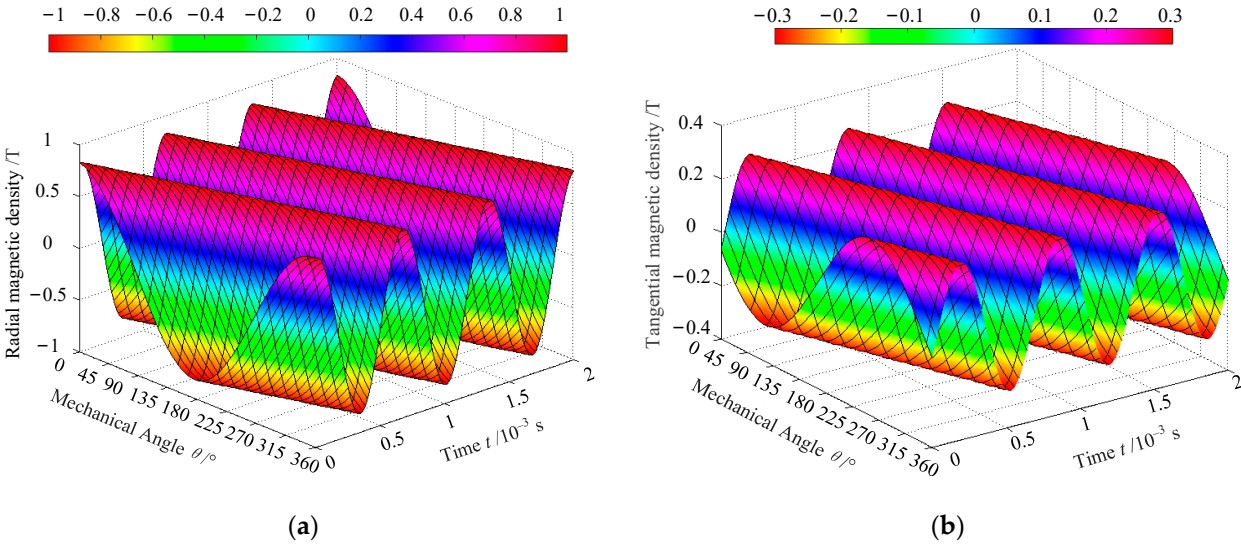

(**a**)                                                                                             (**b**)

**Figure 4.** Distribution of magnetic density along the outer surface of the permanent magnet:
(**a**) Radial magnetic density; (**b**) Tangential magnetic density.

Figures 3 and 4 show that the magnetic density was distributed in a periodic sine
along the direction of the angular axis and time coordinate axis, respectively. The influence
of the stator slot on the magnetic density increased as the proximity to the stator's inner
surface increased.

Based on the air gap flux density in Figures 3 and 4, the radial electromagnetic force
density was obtained by analytical calculation using Equation (7). The radial electromag-
netic force density along the inner surface of the stator and the surface of the permanent
magnet is demonstrated in Figure 5. It was observed that the frequency was twice the
rotation frequency, and the radial electromagnetic force density changed along the angle
and time coordinate axis.

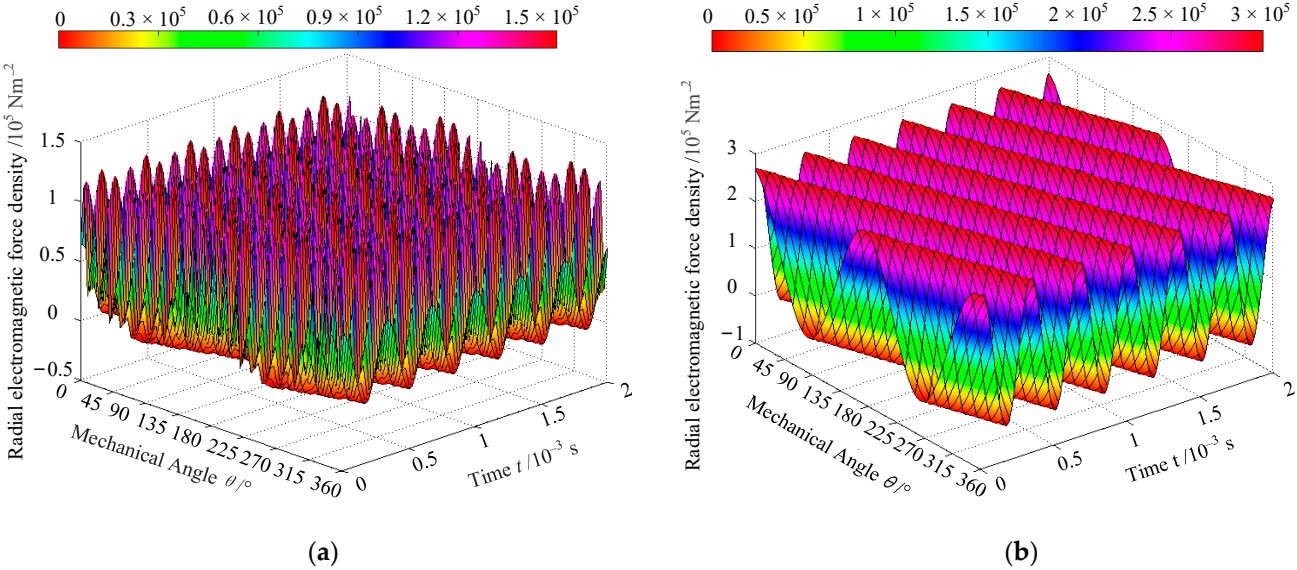

(**a**)                                                                                             (**b**)

**Figure 5.** Results of the radial electromagnetic force density: (**a**) Inner surface of the stator; (**b**) Outer
surface of the permanent magnet.

Figure 6 shows the radial electromagnetic force density on the outer surface of the per-
manent magnet and the inner surface of the stator within one cycle under the load condition.

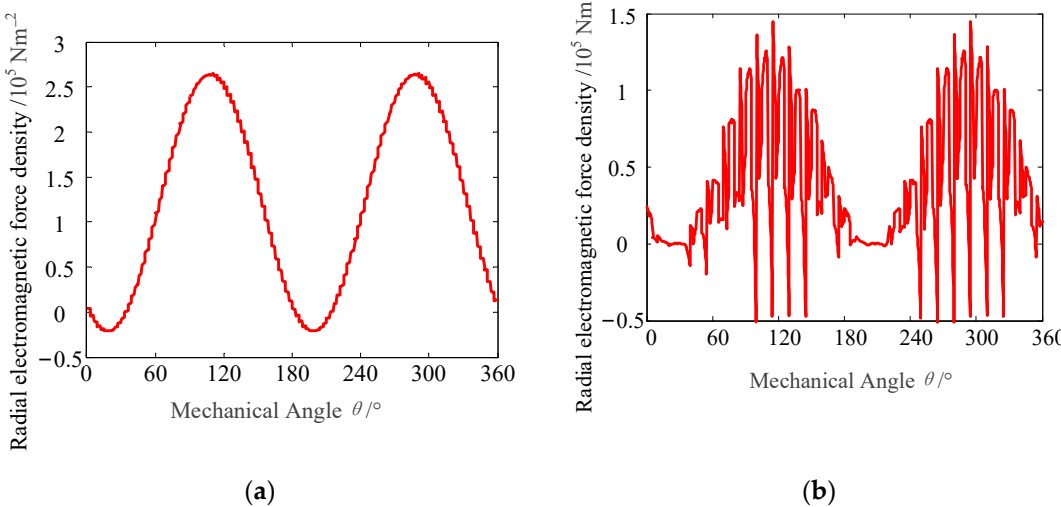

(**a**)          (**b**)

**Figure 6.** Results of the radial electromagnetic force density: (**a**) Outer surface of the permanent magnet; (**b**) Inner surface of the stator.

The harmonics and amplitudes of each order were obtained using the Fourier transform of the radial electromagnetic force density, as shown in Figure 7.

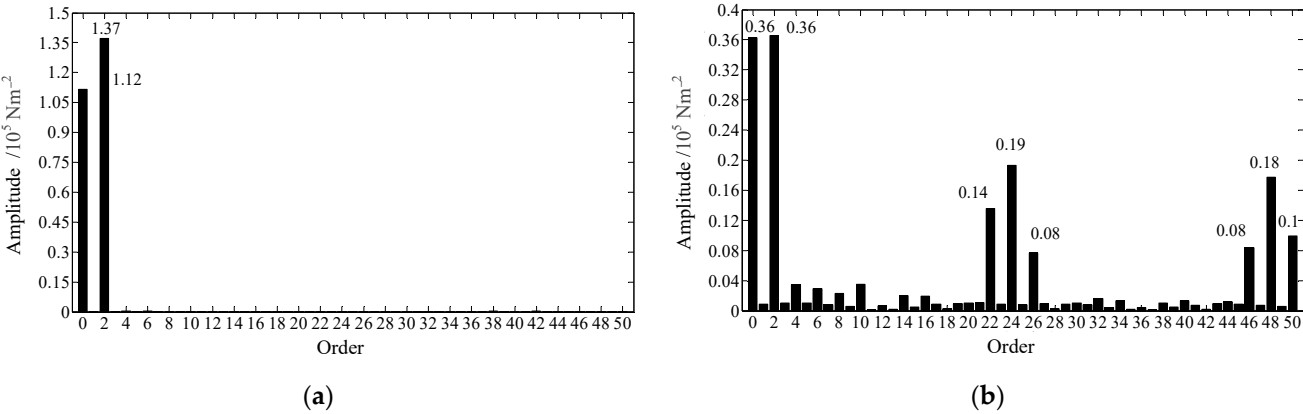

(**a**)          (**b**)

**Figure 7.** Harmonics of the radial electromagnetic force density: (**a**) Outer surface of the permanent magnet; (**b**) Inner surface of the stator.

When $r = R_1$, the main orders on the outer surface of the permanent magnet were 0 and 2, as shown in Figure 7a. The zero-order component was the stationary component. The second-order harmonic component had a large amplitude, and the amplitudes of other harmonic components were very small. The radial electromagnetic force density on the outer surface of the permanent magnet was influenced slightly by the slots of the stator.

When $r = R_2$, the zero-order component was the stationary component on the inner surface of the stator, as shown in Figure 7b. The second-order harmonic component had a large amplitude, and the orders of other harmonic components were all even numbers, which included 22, 24, 26, 46, 48, and 50. The observation from Figure 7b indicated that harmonics repeated cyclically. The radial electromagnetic force density on the inner surface of the stator was influenced by the slots of the stator. Therefore, the orders of harmonic components were related to the number of slots, which was represented by $N_s$ [28]. The generalization of the relationship could be studied and demonstrated in future papers because it was not closely related to the theme of this paper.

The tangential electromagnetic force density on the outer surface of the permanent magnet under the load condition was obtained, as shown in Figure 8. It was seen that both angle and time frequency were twice the rotation frequency.

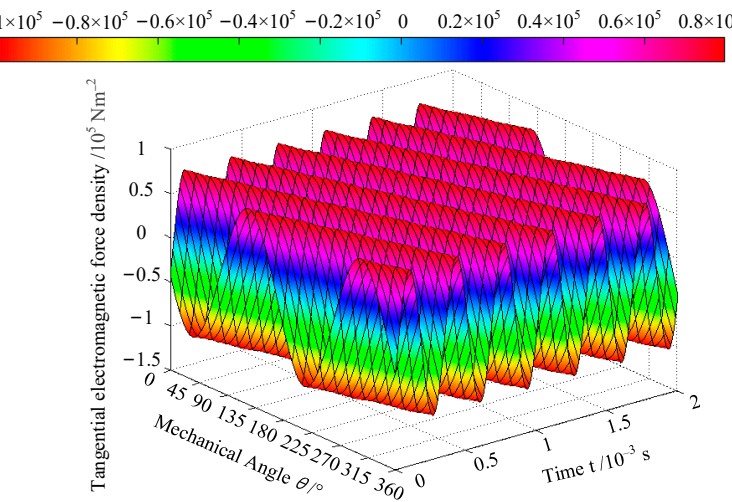

**Figure 8.** Results of the tangential electromagnetic force density.

The characteristic curves of electromagnetic torque and power were calculated using analytical Equation (17), as shown in Figure 9.

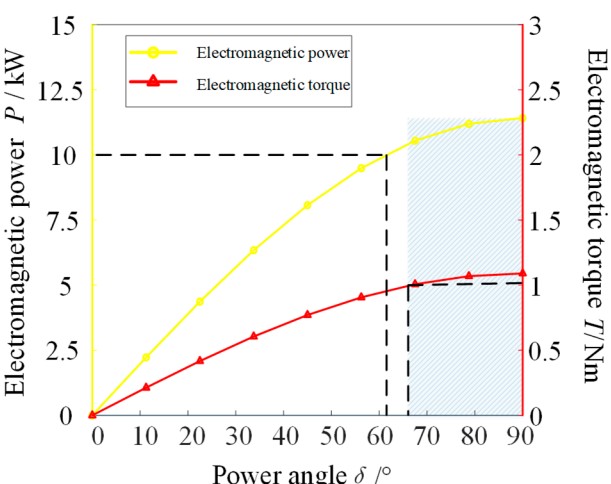

**Figure 9.** Electromagnetic torque and electromagnetic power curves.

When the power angle was 90°, the electromagnetic torque and power had the maximum values. The rated torque of the 10 kW motor was 1.04 Nm. The maximum power was 11.4 kW. The maximum torque was 1.09 Nm, which was 1.05 times greater than the rated torque. It was seen that, on the premise of meeting the electromagnetic power and torque, the allowable range of motor power angle was 66.5°–90°.

By combining the allowable range of motor power angle, the dynamic load characteristics of the compressor system could be verified using Equations (18) and (19).

## 4. Results and Discussion

### 4.1. Electrical Performance

According to the calculated power, the rated current was calculated as follows:

$$I_N = \frac{P_{em}}{\sqrt{3}U_{Nl}\eta_N \cos \varphi_N}, \tag{20}$$

where $U_{Nl}$ is the rated line voltage; $\eta_N$ is the rated efficiency; $\cos\varphi_N$ is the rated power factor.

The no-load-induced voltage was an important parameter. The no-load-induced voltage was calculated as follows:

$$E_0 = 4.44 f K_{dp} N \Phi_1,$$  (21)

where $K_{dp}$ is the winding factor; $N$ is the turns-in-series per phase; $f$ is the rated frequency; $\Phi_1$ is the fundamental magnetic flux of the air gap, which can be calculated using Equations (5) and (6).

The length of the permanent magnet was estimated using the following equation:

$$L_m = \frac{\pi E_0}{2 \times 4.44 f N K_{dp} B_{r1}^{P} \tau},$$  (22)

where $\tau$ is the polar pitch.

Generally speaking, when only performance indicators of PMSMs were required without any other limitations, structural parameters were not unique in the engineering. Therefore, the finite element simulation analysis on the electrical performance of PMSMs needed to be carried out to verify the magnetic density distribution, power, torque, voltage and current. If the electrical performance did not meet requirements, structural parameters were readjusted and analyzed again.

*4.2. Simulation Analysis Results*

The finite element simulation analysis was conducted on the performance of the PMSM. The 2D finite element model was built using the voltage source excitation and considering the length of the stator and the permanent magnet, as shown in Figure 10. Simulation results were obtained by using the software program of ANSYS 18.

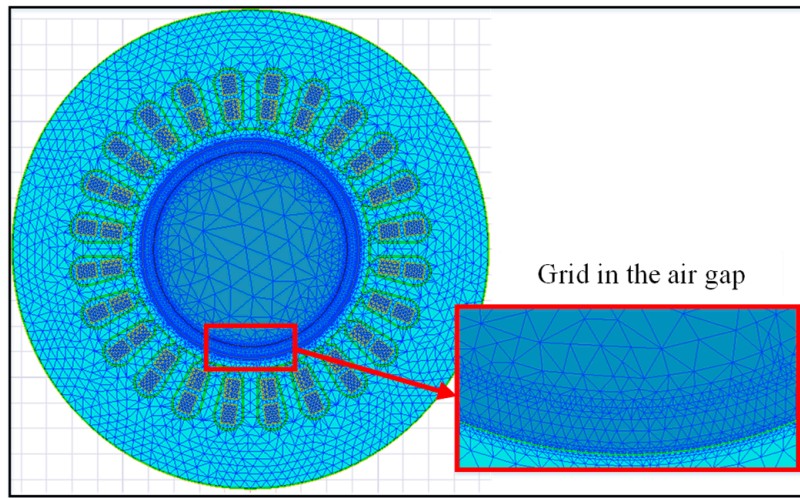

**Figure 10.** Finite element model of the PMSM [29].

Figure 11 shows the no-load induced voltage. The maximum value and root mean square (RMS) value were 286 V and 202.3 V, respectively.

Figure 12 shows the phase current and phase voltage under the load condition. The three-phase current and voltage were symmetrical sine waves. The maximum value and RMS value of phase current were 26 A and 18.4 A, respectively. The maximum value and RMS value of phase voltage were 285.8 V and 202 V, respectively.

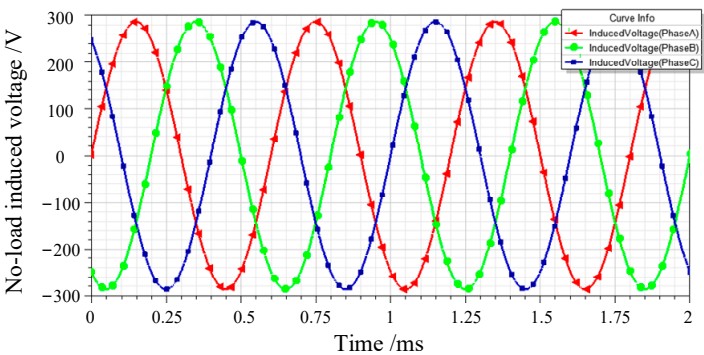

**Figure 11.** No-load induced voltage.

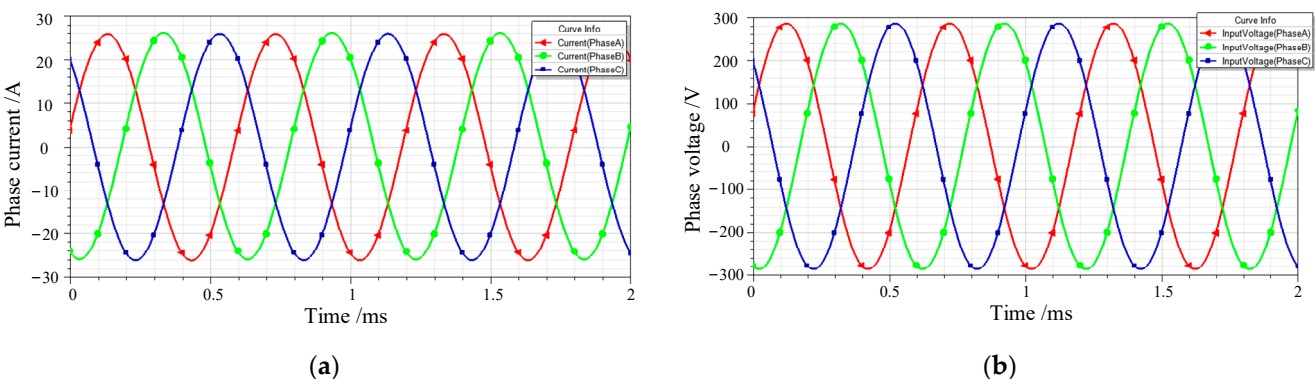

(**a**)                                                    (**b**)

**Figure 12.** Phase current and phase voltage under the load condition: (**a**) Phase current; (**b**) Phase voltage.

Simulation results of magnetic field lines and magnetic density distribution under the load are shown in Figure 13. The maximum magnetic flux density amplitude was 1.57 T. The maximum value of the magnetic density was at the stator teeth, and the value of the magnetic density at the stator yoke was relatively low. The magnetic flux saturation value of the stator core generally did not exceed the requirement of 1.8 T. According to Figure 13, the saturation of the stator core was not reached. The analytical model could not account for the saturation. Therefore, the saturation analysis was carried out through the finite element simulation.

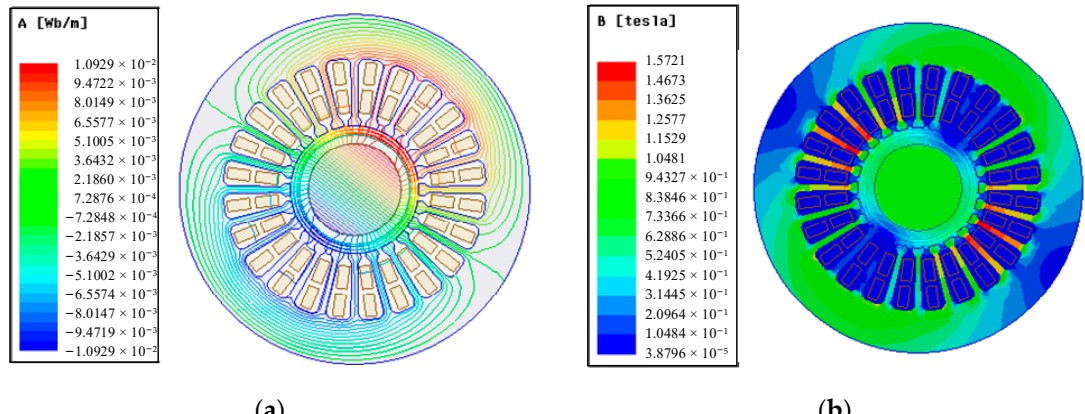

(**a**)                                                    (**b**)

**Figure 13.** Magnetic field lines and magnetic density distribution: (**a**) Magnetic field lines; (**b**) Magnetic density distribution.

The electrical performance parameters of the PMSM are shown in Table 4.

**Table 4.** Electrical performance parameters of the PMSM.

| Parameter | Value | Parameter | Value |
|---|---|---|---|
| Rated power (kW) | 10 | Rated speed (kr/min) | 100 |
| Rated phase voltage (V) | 202 | Rated current (A) | 18.4 |
| No-load induced voltage (V) | 202.3 | Rated torque (Nm) | 1.04 |

*4.3. Description of the Experiment*

The experimental system for high-speed PMSMs that directly drive air compressors is shown in Figure 14. The PMSM was supported by elastic foil gas bearings, and coaxially drove an impeller. The structural parameters of the PMSM are shown in Table 3.

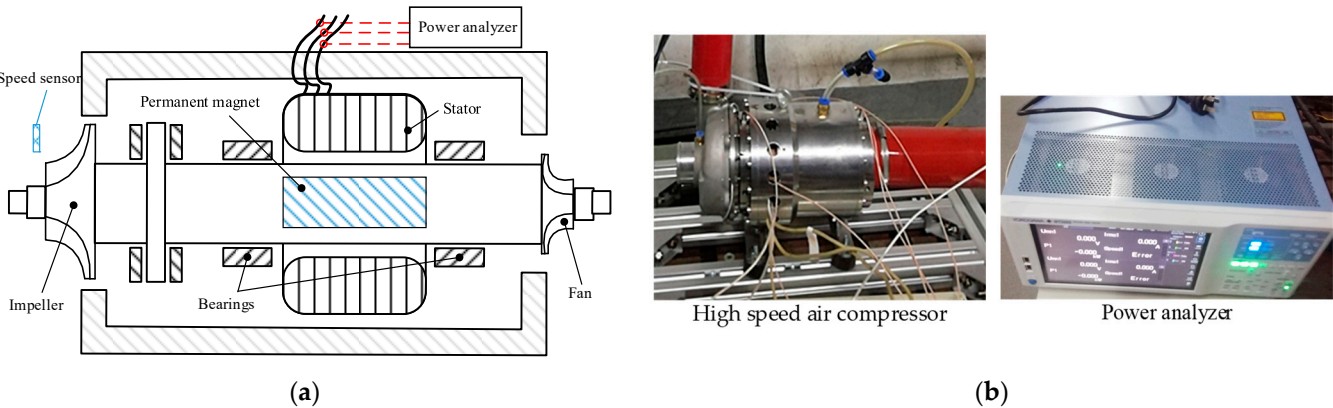

(**a**)          (**b**)

**Figure 14.** Experimental system for high-speed PMSMs: (**a**) Measurement scheme for the acquisition of experimental values; (**b**) Test platform of the high-speed direct-drive air compressor.

The electrical performance experiments of the high-speed PMSM were carried out. Values of voltage, current, power and power factor were tested via the power analyzer. The type of power analyzer used was YOKOGAWA WT5000. The operation rotation was detected by the speed sensor. The type of speed sensor was the MiniVLS laser optical speed sensor. The speed sensor was used with a mounting bracket fixed to the motor surface. This experimental system could determine the motor's rotating speed, torque, and output power. The speed sensor signal was proportional to the motor rotating speed, and the torque meter signal was proportional to the motor torque. Also, the number of motor poles was 2 and the motor's synchronous speed was determined. Furthermore, the active power, frequency, and motor output are measured by this experimental system to compute the motor efficiency.

The maximum experimental operation rotation speed was 90 kr/min. Experimental results of the active power, voltage, current, torque, and power factor were obtained.

*4.4. Experimental Results and Analysis*

Figure 15a shows curves of the power and rotational speed during the operation of the PMSM. When the PMSM operated stably at 90 kr/min, the average power of each phase was approximately 3.12 kW, and the active power was 9.36 kW.

Figure 15b shows curves of the power factor and speed. It was observed that the impeller load and the power factor continuously increased with an increase in speed. It tended to flatten out during stable operation at 90 kr/min. The power factor at 90 kr/min was 0.81, and the power factor angle was 36.4°.

Table 5 provides the experiment values and simulation results of the PMSM during the stable operation at different speeds. The line voltage at 90 kr/min was 321 V, and the line current was 21 A.

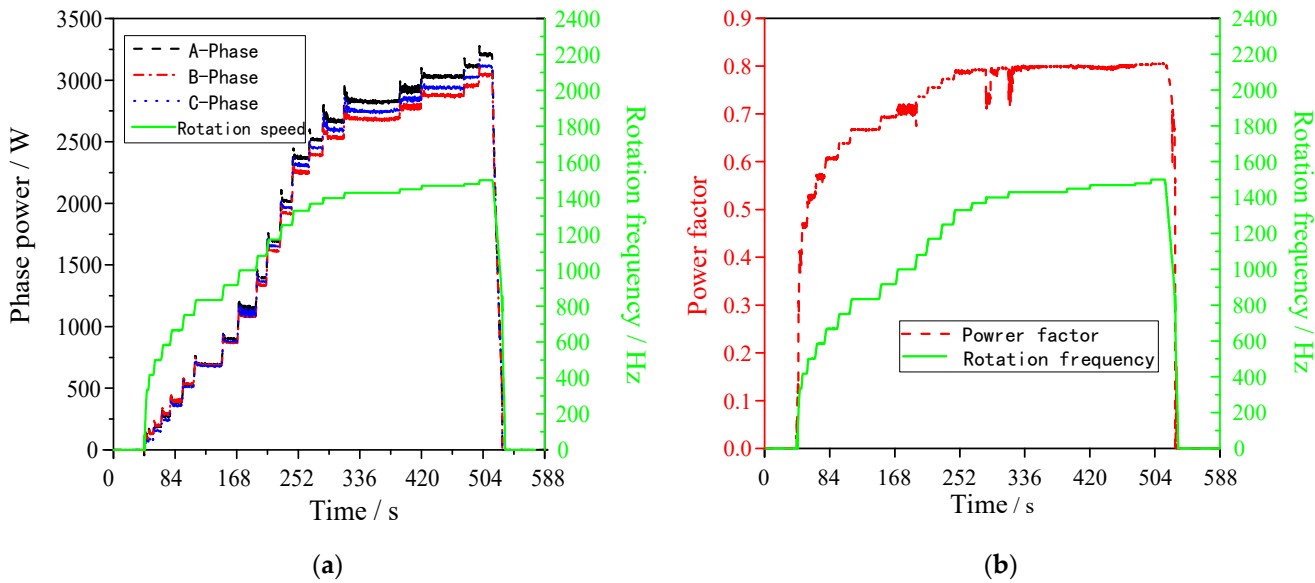

**Figure 15.** Three-phase power and power factor curves with rotation speed: (**a**) Three-phase power; (**b**) Power factor.

**Table 5.** Comparison between experimental values and calculation results.

| Speed (kr/min) | Voltage (V) | Line Current (A) | | | Power Factor | | | Electromagnetic Torque (Nm) | | |
|---|---|---|---|---|---|---|---|---|---|---|
| | | Experiment | Simulation | Deviation | Experiment | Simulation | Deviation | Experiment | Simulation | Deviation |
| 25 | 161.0 | 2.7 | 2.8 | 3.5% | 0.463 | 0.477 | 2.9% | 0.286 | 0.294 | 2.7% |
| 40 | 207.3 | 5.4 | 5.6 | 3.6% | 0.605 | 0.623 | 2.9% | 0.401 | 0.410 | 2.2% |
| 60 | 258.1 | 10.7 | 11.0 | 2.7% | 0.702 | 0.725 | 3.2% | 0.645 | 0.655 | 1.5% |
| 80 | 308.8 | 16.7 | 17.1 | 2.3% | 0.789 | 0.813 | 3.0% | 0.893 | 0.912 | 2.1% |
| 90 | 321.0 | 21.0 | 21.6 | 2.8% | 0.805 | 0.84 | 4% | 0.993 | 1.01 | 1.6% |

The voltage values measured experimentally at different speeds (in Figure 15) were applied as the excitation source, and the performances of the PMSM at different speeds were calculated using Equations (17) and (18). The deviation between calculation results and measured values of the line current, power factor and electromagnetic torque was about 4%.

During the experiment, the impeller did not reach the expected full load operating condition, and the fan did not also achieve the heat dissipation efficiency of the design. The electrical performance experimental results of the PMSM were obtained, and the deviation between the experimental result and the theoretical value was within 4%. Therefore, the analytic expressions could be used in the electrical performance analysis for the high-speed PMSMs.

## 5. Conclusions

The analytic calculation and experimental research conducted in this study focused on developing and establishing an analytic model of the PMSM electromagnetic torque and force density with the goal of providing the theoretical basis and methods for the rotor dynamics analysis. The findings of this study were as follows:

(1) The analytic models of air gap magnetic density, electromagnetic force density, and electromagnetic torque and power were established for PMSMs with a parallel magnetized cylindrical permanent magnet. The electromagnetic force density could be divided into two parts. One part was the stationary component, and the other was the harmonic component.

(2) The analytical model could not account for the saturation and end effect of the stator. This was the weakness and limitation of the analytical approach. The finite element simulation was still necessary to guarantee that the stator saturation did not occur. However,

the analytical solutions of the electromagnetic torque and force were obtained through the analytical model and could be applied in the rotor dynamic equation. It provided the theoretical basis and method for analyzing the influence of electromagnetic load on the rotor vibration. This was the strength of the proposed model.

(3) The analytic calculations of air gap magnetic density, electromagnetic force density, and electromagnetic torque and power were carried out. The harmonics and amplitudes of the electromagnetic force density were obtained. The zero-order component was the stationary component. The second-order harmonic component had a large amplitude. The relationship between the harmonic component and the number of slots needs to be studied and demonstrated in future papers.

(4) The finite element simulation analysis on the electrical performance of the PMSM was carried out to verify magnetic density distribution, power, torque, voltage and current. The maximum amplitude of magnetic flux density was 1.57 T, which is located at the stator teeth. The electrical performance meets the design requirement.

(5) The electrical performance experiment for high-speed PMSMs that directly drive air compressors was carried out. The deviation between the experimental result and the theoretical value was within 4%. This result verified that the analytic expressions could be used for application in engineering and industrial contexts.

**Author Contributions:** Conceptualization, H.L. and H.G.; methodology, H.L.; software, X.H.; validation, H.L.; formal analysis, H.L.; investigation, L.L.; resources, H.L.; data curation, H.G.; writing—original draft preparation, H.L.; writing—review and editing, H.L. and L.S.; visualization, H.L.; supervision, H.L.; project administration, H.L.; funding acquisition, H.L. All authors have read and agreed to the published version of the manuscript.

**Funding:** This research was funded by the Fundamental Research Funds for the Central Universities, grant number 2022JBZY028. This research was supported by Beijing Natural Science Foundation, grant number 3244035.

**Data Availability Statement:** The data are contained within the article.

**Conflicts of Interest:** The authors declare no conflicts of interest.

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
