# Peer review of "Analytical Solution for Electromagnetic Performance Analysis of Permanent Magnet Synchronous Motor with a Parallel Magnetized Cylindrical Permanent Magnet"

_machines, doi:10.3390/machines12030152_

Round 1
Reviewer 1 Report
Comments and Suggestions for Authors
Authors presented the analytical model of the air gap magnetic density, electromagnetic force density and electromagnetic performance for high-speed PMSM with a parallel magnetized cylindrical permanent magnet.
MAIN COMMENTS:
1.) It is not clear what is new. Are the equations of the analytical approach developed by authors or are these equations taken from the literature? Write clearly in the introduction what the scientific contributions of the article are (it is best if you list it in points).
2.) Not only current and power factor, also at least torque should be compared with experimental model.
3.) Clearly write the strengths of the proposed model, the weaknesses of the proposed model and the limitations of the proposed model.
4.) Clearly write what is different in your model than in existing models
DETAILED COMMENTS:
Line 23: 4% is low value for motor design considering difference between analytical and experimental results.
Line 103: Please write clearly what the scientific contributions of this article are. It is best if you list it in points.
Line 113: Considering assumptions (ignoring end effect, stator permeability is considered as infinite, permanent magnet has a linear demagnetization curve …) it is difficult to believe to get only 4% deviation between analytic and experimental model.
Line 120: What is the meaning of “parallel magnetization” written in the Figure 1? Magnetization in the air gap is radial to my knowledge.
Line 126: all values used in equations (1), (2), (3), (4), (5) and (6) should be explained (for better transparency, they can also be presented in a table)
Line 126: I assume that equations (1)-(6) are taken from the literature, please provide it
Line 142: All values used in the equations (7) – (14) should be explained (for better transparency, they cen also be presented in a table). If they are taken form the literature, please provide it.
Line 172: All values used in the equations (15) – (19) should be explained (for better transparency, they cen also be presented in a table). If they are taken form the literature, please provide it.
Line 198: I'm used to units being written in brackets. If there is no requirement to write as you wrote, correct in brackets. For example Br/T change to Br (T). Please consider that for all units used in the paper (disregard if the magazine's requirement is to write your way)
Line 198: What is the length of the tested motor? The end effect is neglected and it is necessary to specify the length of the motor.
Line 214: It is written “… electromagnetic force was obtained by numerical calculation using Equation (7). Why numerical?
Line 270: 2D finite element model was built for verification. In 2D model there are similar restrictions. It will be much better to build 3D model and than compared analytical with FEM model. Why these analysis were conducted, because at the end of the paper the experimental system is presented for verification.
Line 289: According to figure 13 saturation is not reached. I assume that the analytical model cannot account for saturation. Please write this clearly in the article as a weakness of the analytical approach.
Line 319: Only line current and power factor were compared. Could you compare calculated and measured torque of the motor? Even if current and power factor are similar, it does not mean that other important quantities are similar (such as torque).
Author Response
Thank you very much for taking the time to review this manuscript. Please see the attachment, which is the response to the reviewer’s comments.

Reviewer 2 Report
Comments and Suggestions for Authors
The manuscript „Analytical solution for electromagnetic performance of a high speed PMSM with parallel magnetized cylindrical permanent magnet” presents a study analyzing the electromagnetic machine properties.
The manuscript is well structured and impresses with its good readability. The presented analysis is a well worked out example of an analytic approach to machine calculation. However, in my opinion it fails to reach the intended goal of a presenting a model that improves design accuracy. Please consider my remarks
1) Several analytical models for PMSM are well known and the authors present a good scoping review. The presented example with a high number of slots and a diametrally magnetized PM is one of the best examples for an analytical approach. Even if the authors could not find the exact example in literature, it can be easily deducted from the examples found in literature.
2) I could not find any arguments why the analytical model should outperform the FE solution. No result that could not be derived with the FEM analysis is presented.
3) The analytical model does not consider saturation. The FEM solution reveals a max. flux density close to the saturation region. It is a coincidence that no severe saturation occurs and not a feature of the model. In the overload range the analytical model will diverge from the FEM solution. Moreover, 2D FE is easy to implement and simulates fast.
4) Concerning the presentation of the equations:
a. Please explain all symbols directly after their first use.
b. Show the coordinate systems used and explain if they are stator or rotor fixed.
c. Indicate the point r=0 or r running from R1 to R2
d. Better explain how equation 2 is derived and illustrate all angles delta, beta, gamma, theta and phi in the figure
5) The model does not include specific high speed losses. Moreover, not even ohmic losses are mentioned. The fact, that the results influenced by the losses are in good accordance with the simulation results from the efficiency introduced in (20). It remains unclear how this value is defined. Tuning this factor will allow any value of accordance between model and experiment. I my opinion, it contradicts the argument to improve the design accuracy.
Comments on the Quality of English Language
The manuscript is well structured and impresses with its good readability.
Author Response

(The authors gave the same response as above.)

Reviewer 3 Report
Comments and Suggestions for Authors
1. most of the theoretical formulas, figures 1, 2, 10, have no references; many of the quantities used are not defined; Please fix the situation.
2. specify the nominal parameters of the studied electric motor.
3. specify the software (respectively the version) used for the graphic presentation and numerical analysis.
4. in fig.1, the positioning in the air, in the upper left of the electric machine, of "parallel magnetization" can create controversy.
5. from figure 2 it does not appear that the studied electric car has NS=24.
6. in the tables, the units of measure must be written in parentheses.
7. I don't see the need to specify the universal constant from table 1, nor the multiplication by 1000 from (20).
8. draw / specify the measurement scheme used for the acquisition of experimental values and describe the working method; also specify the type of devices used (analyzer and sensor). Otherwise, the photos presented are irrelevant.
9. the conclusion in which it is suggested that the presence of harmonics of a certain rank seems to repeat itself according to a certain rule must be scientifically supported, not just affirmed.
10. the conclusions are irrelevant and need to be improved.
Comments on the Quality of English Language
Paper is written in good scientific English.
Author Response

(The authors gave the same response as above.)

Reviewer 4 Report
Comments and Suggestions for Authors
Was a great pleasure for me to read you articole. It’s a very good article, very well written and structured. You starts with developing analytical models and next the analytical calculation results are validated experimental.
I have only one suggestion for you: please mention the software program that you used to obtain the simulations results (for example, as those from FIgures 13).
I congratulate you for the analytical models and for the very good analitycal and experimental results obtained in your research activities described in this articole. Your models will be very usefull for the researches and enginners in this domain.
Author Response

(The authors gave the same response as above.)

Round 2
Reviewer 1 Report
Comments and Suggestions for Authors
Avtors considered all my comments. Paper can be published in the present form.
Author Response
Thank you very much for taking the time to review this manuscript.
Reviewer 2 Report
Comments and Suggestions for Authors
The revision of the manuscript with the title “Analytical solution for electromagnetic performance of a high-speed PMSM with a parallel magnetized cylindrical permanent magnet“ has improved in presentation and the examination of symbols. This has improved readability.
However, the main points of criticism have not be addressed with the required depth.
1) The paper still uses several general terms. I just list some here for example – please revise the whole text
a. “Some primary structural (…) parameters…” – but it is completely unclear which? Moreover, you present an analysis (and this is currently stated in the title) of a design and not a layout or an optimization. There is no need to identify some structural parameters, because all parameters are given, which is required to set up the analytic model. Thus, only performance parameters remain. But theses are based in both cases (analytical and FEM) on the post processing. I simply do not see the point. Please discuss these “some (..) parameters” in detail. A simple list will not be sufficing.
b. “improved design efficiency” – What is “design efficiency”? How is it defined and measured? How could it be improved in contrast to which situation. To be clear: The speed and accuracy of design has definitely not been improved. 2D-FEM Analysis is a matter of minutes and it is easy to implement. The given machines design is a standard machine design without any special properties.
2) The paper promises a “analytic solution for (…) a high-speed …” but not a single aspect of high speed is considered in the paper. Your answer to my comment (5) is more or less a copy&paste of your answer to my comment (3). I am not satisfied with that and disagree to your argument that high-speed is more or less a matter of saturation. Even if it would, you would have to expand your model to consider it.
3) You mention that FEM is still needed to check if the linear model is valid. Thus, you’ll need Fem anyway and FEM+analytic is definitely less efficient in design that FEM alone.
4) Concerning conclusion (4) and (5): How can the power be derived and verified in FEM when the efficiency is only measured? Please don’t mix up analysis and design! The verified analysis of one machine does not give an argument for the design. Simple example: Saturation could occur.
Comments on the Quality of English Language
easiliy readable
Author Response

(The authors gave the same response as above.)

Reviewer 3 Report
Comments and Suggestions for Authors
I appreciate the effort put into improving the paper. At this point, the paper looks much better.
However, there is one more conclusion that is not supported: to clarify the conclusion, according to which there are important harmonics that repeat cyclically according to the relationships described in the paper on lines 273 and 419 respectively. The generalization of the observation from fig. 7b is not supported in any way; additional arguments must be added to demonstrate its validity.
Author Response

(The authors gave the same response as above.)

Round 3
Reviewer 2 Report
Comments and Suggestions for Authors
The 2nd revision of the manuscript with the title “Analytical solution for electromagnetic performance analysis with a parallel magnetized cylindrical permanent magnet“ has not improved the depth of the content, but highlighted the significant limits of the proposed approach. Taking out “high speed” from the title was an important step to bring content and title together.
However, the main question
1) What is the worth of a model, that cannot consider saturation?
remains. Please add that a FEM simulation is still necessary to guarantee that saturation does not occur and that the intention of this modelling is not to find a replacement for the FEM approach, but an accompanying analytic solution that can be beneficial for further analytic mechanical modelling and analysis, e.g the mentioned rotor dynamics.
Author Response

(The authors gave the same response as above.)
